# Usability of EU-TIRADS in the Diagnostics of Hürthle Cell Thyroid Nodules with Equivocal Cytology

**DOI:** 10.3390/jcm9113410

**Published:** 2020-10-24

**Authors:** Dorota Słowińska-Klencka, Kamila Wysocka-Konieczna, Mariusz Klencki, Bożena Popowicz

**Affiliations:** Department of Morphometry of Endocrine Glands, Chair of Endocrinology, Medical University of Lodz, Pomorska Str 251, 92-213 Łódź, Poland; kamila.wysocka91@wp.pl (K.W.-K.); marklen@tyreo.umed.lodz.pl (M.K.); bozena.popowicz@umed.lodz.pl (B.P.)

**Keywords:** EU-TIRADS, FNA, thyroid ultrasonography, Hürthle cell, thyroid cancer

## Abstract

The aim of this study was to compare the diagnostic effectiveness of EU-TIRADS in two groups of nodules with equivocal cytology (categories III-V of Bethesda system), with and without Hürthle cells (HC and non-HC). The study included 162 HC and 378 non-HC nodules with determined histopathological diagnosis (17.9% and 15.6% cancers). In both groups calculated and expected risk of malignancy (RoM) for high, intermediate and benign risk categories of EU-TIRADS were concordant. RoM for low risk category was higher than expected in both groups, but especially in HC (HC: 13.9%, non-HC: 7.0%, expected: 2–4%). The majority of cancers in HC of that category were follicular thyroid carcinomas (FTC) and Hürthle cell thyroid carcinoma (HTC) (60.0% vs. non-HC: 16.7%). The diagnostic efficacy of EU-TIRADS was lower in HC (the area under the receiver operating characteristics curve (AUC): 0.621, sensitivity (SEN): 44.8%, specificity (SPC): 78.9% for high risk threshold) than in non-HC (AUC: 0.711, SEN: 61.0%, SPC: 77.7%). AUC was the highest for category V (AUC > 0.8, both groups) and the lowest for category IV (inefficient, both group). If intermediate risk category was interpreted as an indication for surgery, 25% of cancers from category III and 21.4% from category IV would not be treated in the HC group (0.0% and 7.4% from non-HC group, respectively). EU-TIRADS does not aid making clinical decisions in patients with cytologically equivocal HC nodules, particularly those classified into category IV of Bethesda System for Reporting Thyroid Cytopathology (BSRTC).

## 1. Introduction

The Hürthle cells (HC) are cells with features characteristic of an oncocyte, which are found in the thyroid gland. These features include abundant, finely granular cytoplasm (as a result of the presence of innumerable large mitochondria), a round nucleus and a variably conspicuous nucleolus [1]. The HC may constitute foci or may be scattered. Neoplastic nodules composed of at least 75% of HC correspond to Hürthle cell neoplasms (HN), adenomas (HTA) or carcinomas (HTC). These cells may also appear in follicular thyroid adenomas (FTA), follicular thyroid carcinomas (FTC) and papillary thyroid carcinomas (PTC). Scattered HC are also present in non-neoplastic lesions, nodular goiter (especially in the elderly), Hashimoto disease, Graves’ disease, lesions induced by radiotherapy or systemic chemotherapy [1].

The preoperative diagnostics of HC lesions with the use of fine-needle aspiration biopsy (FNA) is difficult. Smears containing HC may be classified into each of the six categories of the Bethesda System for Reporting Thyroid Cytopathology (BSRTC) depending on the microscopic image [2,3]. However, both our analyses as well as studies by Yazgan et al. indicate that smears with HC are classified into categories of equivocal cytological outcomes (categories III–V) more often than smears without HC [4,5]. There are no unequivocal criteria for distinguishing HTC from HTA. Similarly to follicular neoplasms (FN), the FNA outcome in these cases is usually classified into category IV: suspicion of follicular neoplasm (SFN)/suspicion of Hürthle cell tumor (SHCT) or category III: follicular lesion of undetermined significance (FLUS)/atypia of undetermined significance (AUS) of the Bethesda System [2,3]. Moreover, nodules with HC are quite often classified into category V due to the resemblance between HC and PTC cells, which usually show oncocyte-like oxyphilic cytoplasm [1]. Category V is an indication for surgical treatment and the risk of malignancy associated with this category exceeds 50%. Category IV is usually also regarded as an indication for surgical treatment, but in this case the surgery may be forgone in areas of endemic goiter, which is justified by relatively low risk of malignancy (in such areas nodules of category IV most often correspond to non-neoplastic hyperplastic nodules). Among surgically excised nodules reported as SFN/SHCT the malignancy rate is 10–40% [3]. Category III is not an indication for surgical treatment but for performing control FNA and molecular tests if available. As the classification of smears into category III is highly variable, the estimated risk of malignancy varies from several up to nearly 70% [3,6].

Frequent classification of nodules with HC into equivocal diagnostic categories stimulates the search for additional methods that could facilitate making clinical decisions in patients with such nodules. One of these methods is the evaluation of a ultrasonographic image of a nodule. Currently, it is usually performed with the use of ultrasonographic risk stratification systems (U-RSS), commonly called thyroid imaging reporting and data systems (TIRADS) [7,8,9,10,11,12]. The distinguishing feature of these systems is the definition of several classes of ultrasonographic images of nodules that differ in estimated risk of malignancy (RoM). U-RSS are optimized to reveal the most common thyroid cancers (i.e., PTC) and are less efficient in the case of other cancers, particularly FTC and HTC. Nevertheless, our earlier comparative analysis of six U-RSS showed that all evaluated systems aided in the selection of FLUS/AUS nodules for surgical treatment in a population characterized by a low risk of malignancy in nodules with indeterminate cytology and by a low percentage of PTC among cancers [13]. This analysis included five U-RSS recommended by relevant scientific societies: European Thyroid Association (EU-TIRADS), Korean Society of Thyroid Radiology (K-TIRADS), American College of Radiology (ACR-TIRADS), the American Thyroid Association (ATA) guidelines, the American Association of Clinical Endocrinologists/American College of Endocrinology-Associazione Medici Endocrinologi (AACE/ACE/AME) guidelines and the system developed by Kwak. Among the analyzed systems, EU-TIRADS showed the greatest versatility in relation to various types of cancers. Similarly, Castellana et al. (2020) indicated that EU-TIRADS was characterized by the low frequency of missed FTC in comparison to other U-RSS [14]. Due to this, in the present study, we decided to assess whether EU-TIRADS is also effective in a very specific group of nodules with equivocal cytology, i.e., nodules with HC. The aim of the study was to compare the diagnostic efficacy of EU-TIRADS in two groups of cytologically equivocal nodules—with and without HC—as well as to evaluate the usefulness of that system in determining the need for surgical treatment.

## 2. Materials and Methods

### 2.1. Examined Patients

The study was made in a single center where thyroid FNA and ultrasound imaging were performed in patients referred by endocrinologists from outpatient clinics. The analysis included patients with thyroid nodules subjected to FNA in the years 2010–2019 with the outcome conforming to the categories III, IV or V in the Bethesda classification, who subsequently underwent thyroid surgery. Exclusion criteria were: (a) lack of full ultrasound imaging data, (b) lack of information on the results of the postoperative histopathological examination, (c) previous surgical or radioiodine thyroid treatment, (d) positive neck irradiation history. All nodules included in the study were divided into two groups depending on the presence of HC in a smear (HC and non-HC groups). Overall, the study included 162 HC nodules revealed in 138 patients and 378 non-HC nodules revealed in 347 patients (Table 1).

Patients with a cytological outcome of SFN/SHCT or suspicion of malignancy (SM) were routinely referred for surgical treatment. In the case of a diagnosis of FLUS/AUS, the surgical treatment was performed based on the patient’s preference or due to the large size of the goiter, as well as the presence of other clinical risk features. The histopathological examination revealed cancers in 17.9% of HC nodules and 15.6% of non-HC nodules (NS). Cancers among both HC and non-HC nodules were more frequent in those classified into category V of BSRTC rather than those classified into categories III or IV (Table 1). The percentage of PTC among cancers was significantly lower in the group of HC nodules (44.8%) than in non-HC nodules (74.6%, *p* < 0.01). PTCs were the most frequent cancers among nodules classified into category V of BSRTC in both groups (HC: 85.7%, non-HC: 92.6%) and in category III in the non-HC group (63.0%). The incidence of PTCs was under 40% in other categories of BSRTC in both groups.

### 2.2. Microscopic Examination

The biopsy was performed on thyroid nodules with a diameter of at least 5 mm (and usually over 1 cm) and at least one malignancy risk factor (ultrasonographic or clinical). In most cases, two aspirations of a nodule were performed. Smears were fixed with 95% ethanol solution and stained with hematoxylin and eosin. A detailed description of the classification of nodules into specific diagnostic categories of the Bethesda system was presented in our earlier report [4].

The histopathologic examination was performed according to the standard procedure and its results were formulated according to the WHO classification of thyroid tumors that was in effect at the time of examination. The reclassification of the histopathological examination in order to reveal cases of non-invasive follicular thyroid neoplasm with papillary-like nuclear features (NIFTP) was not performed, similarly to our previous study [13].

### 2.3. Analysis of US Malignancy Features

Analysis of ultrasonographic (US) malignancy risk features was done prospectively directly before FNA. The presence of particular features was assessed by experienced sonographers (doctors with minimum of ten years of experience), according to a unified pattern. We used a computer system dedicated for collecting detailed information on examined nodules in a database. On the basis of these data, the presence of sonographic features relevant for EU-TIRADS were determined: (1) marked hypoechogenicity: darker than the surrounding strap muscles; (2) hypoechogenicity as compared to the normal thyroid (in both cases the lowest echogenicity irrespective of its volume share are assigned); (3) suspicious shape: non oval (taller than wide or round); (4) irregular margins including microlobulated, spiculated and suggesting extrathyroidal extension; (5) microcalcifications: calcifications around 1 mm in size without posterior shadowing located in the solid component of a nodule); (6) spongiform composition involving the entire nodule); (7) pure cystic echostructure composed entirely or nearly entirely of liquid. The presence of other ultrasound features was assessed as follows: (8) solid echostructure (>90% solid); (9) more solid than cystic echostructure (>50% solid); (10) macrocalcifications; (11) rim calcifications; (12) pathological vascularization (marked intranodular vascular spots).

The US examinations were performed with the use of the Aloka Prosound Alpha 7 ultrasound system (ALOKA co. Ltd., Tokyo, Japan) with a 7.5–14 MHz linear transducer. With the use of the set of above-mentioned features, all thyroid nodules were classified into specific categories of EU-TIRADS [9]. The system defines 5 categories. EU-TIRADS 1 refers to a US examination where no thyroid nodule is found. EU-TIRADS 2 (benign category) includes two patterns: pure/anechoic cysts and entirely spongiform nodules. EU-TIRADS 3 (low-risk category) includes isoechoic or hyperechoic nodules with an oval shape, smooth margins and without any feature of high risk of malignancy. EU-TIRADS 4 (intermediate-risk category) includes mildly hypoechoic nodules with an oval shape, smooth margins and without any feature of high risk. In the case of heterogeneous echogenicity of the solid component, the presence of any hypoechoic tissue classifies the nodule into intermediate risk. EU-TIRADS 5 (high-risk category) includes nodules with at least 1 of the following high-risk features: non-oval shape, irregular margins, microcalcifications, and marked hypoechogenicity. Two researchers (KWK and DSK) independently assigned all the ultrasound features for TIRADS score calculation. In the case of discrepancy, the US report was jointly reevaluated and discussed to confirm its categorization.

### 2.4. Analyses, Statistical Evaluation

The incidence of US malignancy features was assessed in the HC and non-HC groups and in the HC and non-HC nodules classified into particular diagnostic categories of FNA (III, IV and V) in respect of the division of the nodules into benign lesions and cancers in the postoperative histopathological examination. The associations between individual US malignancy features and malignancy were evaluated with the use of logistic regression analysis in HC and non-HC groups. Odds ratios (OR) with relative 95% confidence intervals (95% CI) were calculated to determine the relevance of all potential predictors of the outcome. Next, the distribution of benign and malignant nodules among particular categories of the EU-TIRADS was assessed. This led to the determination of RoM for nodules in each of the EU-TIRADS categories—T RoM (the proportion of cancers among all nodules in each category)—in the entire examined set of nodules and in relation to the FNA result. We calculated how T-RoM of a nodule influenced its RoM related to the class of the FNA outcome (FNA-RoM) in the same way as described in our previous study [13]. Next, the threshold category of EU-TIRADS that showed the highest efficiency in the classification of benign and malign lesions was identified by the analysis of the receiver operating characteristics curve (ROC) and the area under the ROC (AUC) value. The effectiveness of the determined threshold in both groups was presented as the sensitivity (SEN), the specificity (SPC), the accuracy (ACC), the negative predictive value (PPV) and the negative predictive value (NPV). The odds ratio (OR) for the established cut-off value was assessed with the use of logistic regression analysis. The same calculations were also performed for the cut-off value set one category lower. Finally, it was evaluated how various cut-off categories in EU-TIRADS would affect the optimization of the number of thyroid surgeries in respect to cancer treatment in patients with nodules of categories III and IV of BSRTC. The statistical analysis was performed with Statistica, version 10 statistical software. The comparison of frequency distributions was performed with chi2 test (with modifications appropriate for the number of analyzed cases). The Kruskal–Wallis test was used for comparing continuous variables between groups. The value of 0.05 was assumed as the level of significance. The study design was approved by the Local Bioethics Committee at the Maria Sklodowska-Curie National Research Institute of Oncology, Warsaw, Poland on 21.06.2016—as a part of the studies carried out for the realization of the grant MILESTONE. The approval code is “13/2015/1/2016”. All patients gave their informed consent.

## 3. Results

Table 2 shows data on the incidence of evaluated US features in HC and non-HC nodules with respect to the results of the postoperative histopathological examination. The univariate logistic regression analysis confirmed that marked hypoechogenicity, irregular margins and microcalcifications were each relevant for predicting malignancy in both HC and non-HC nodules. In the non-HC group, the presence of solid echostructure and hypoechogenicity was also relevant for diagnosing cancer. In the HC group, the solid echostructure was slightly more frequent in benign nodules than in cancers (benign nodules with HC were solid more often than benign nodules without HC, *p* < 0.001). Multivariate logistic regression analysis showed that only irregular margins were an independent malignancy risk feature in both examined groups of nodules: HC (OR: 5.5, CI95%: 1.0–29.0, *p* < 0.05) and non-HC (OR: 8.0, CI95%: 3.5–17.5, *p* < 0.0001). In the HC group, the marked hypoechogenicity was also such a factor (OR: 6.6, CI95%: 1.9–23.1, *p* < 0.005). Detailed analysis of the incidence of evaluated US features, in particular categories of cytological diagnoses, is presented in Appendix A in the supplementary material. In the case of category III of BSRTC, in the HC group the irregular margins were more often identified in cancers than in benign nodules, while in the non-HC group an analogous difference was found for microcalcifications. In the case of category IV, the irregular margins were found only in one of the five cancers in the non-HC group and in none of the 14 cancers in the HC group. The latter group of cancers showed marked hypoechogenicity more often than benign nodules with HC of category IV. Among nodules of category V, all US malignancy risk features except for hypoechogenicity, pathological vascularization, solid and more solid than cystic echostructure appeared only in cancers, but the differences in their incidence between cancers and benign nodules were not statistically significant.

Table 3 shows the distribution of benign and malignant nodules between particular categories of EU-TIRADS and the comparison of calculated and expected T-RoM in these categories. The distribution of benign nodules to particular categories of EU-TIRADS was similar in both examined groups. Noticeable, although still insignificant, differences were identified for malignant nodules: cancers of the HC group were more often categorized to low risk (17.2% vs. 10.2) and intermediate risk (37.9% vs. 28.8%) categories, and less often to the high risk category (44.8% vs. 61.0%) than cancers of the non-HC group. In both groups, the percentage of cancers in the low risk category was higher than expected (2–4%), but the calculated T-RoM for that category was almost twice as high in the HC group as in the non-HC group (13.9% vs. 7.0%). In the case of other categories, the calculated T-RoM was concordant with expectations. In the HC group, the majority of cancers in the low risk category were FTC and HTC: in total 60.0% vs. 16.7% in non-HC group (NS). In both groups, the majority of PTCs were classified into the highest risk category (HC: 61.5%, non-HC 63.6%), while the majority of FTCs and HTCs were located in the categories of intermediate risk (HC: 50.0% and 45.4%; non-HC: 50.0% and 100.0%, respectively).

Appendix A in the supplementary material shows the distribution of benign and malignant nodules between particular categories of EU-TIRADS with regard to particular categories of cytological diagnoses according to the BSRTC. In both groups, the assignment of a nodule to the high risk category of EU-TIRADS was related to the RoM of the nodule being higher than its initial FNA-RoM, but significant differences were observed only in FLUS/AUS non-HC nodules. In the case of SM nodules, the assignment of a nodule to the mentioned category increased the RoM to 100% in both groups.

The diagnostic efficacy of the evaluated system, as measured by AUC, was weak in the HC group (AUC: 0.621), and lower than in the non-HC group (AUC: 0.711) (Table 4). The lower the percentage of PTC among cancers, the lower the AUC values in both groups. The values were highest for nodules classified into category V (AUC > 0.8 in both groups), and lowest for nodules of category IV. For the latter, EU-TIRADS was inefficient in distinguishing between benign and malignant nodules. In the case of category III, the system was efficient only for non-HC nodules. The classification of a FLUS/AUS nodule of the non-HC group into the high-risk category or intermediate risk category of EU-TIRADS significantly increased the risk of malignancy (OR > 5 for both categories). In the case of FLUS/AUS nodules of the HC group, the increased OR was close to significant only when a nodule was classified into the high-risk category of EU-TIRADS.

For the high risk category, which proved to serve as a threshold characterized by the highest values of ACC in both groups, SEN was 15 percentage points lower in the HC group than in non-HC group (44.8% vs. 61.0%), while SPC was comparable (HC: 78.9%, non-HC: 77.7%). When the cut-off value was set at the category of one grade lower risk (intermediate risk), the increase in SEN was observed in both groups, and it was higher in the HC group than the non-HC group (HC: 82.8%, non-HC: 89.8%) while SPC markedly decreased (HC: 24.8%, non-HC: 28.2%).

In the case of FLUS/AUS nodules, the consideration of the intermediate risk or higher category of EU-TIRDAS as an indication for surgical treatment would result in thyroidectomy performed for 67.9% of HC nodules including 75.0% cancers and for 72.8% of non-HC nodules including 92.6% cancers. If all FLUS/AUS nodules of the low risk category of EU-TIRADS were also treated surgically, it would allow the excising of 100% of cancers in both groups, but it would require performing surgery in 98.1% of HC nodules and 96.7% of non-HC nodules.

On the other hand, refraining from surgery in the case of nodules of category IV BSRTC with benign or low risk classification in EU-TIRADS would concern 19.6% of HC nodules including 21.4% of cancers and 18.7% of non-HC nodules including none with cancer. Setting the threshold at the benign category in the HC group would result in the lack of possibility to limit the surgical treatment.

## 4. Discussion

Nodules with HC present in smears constitute less than 10% of all nodules examined with FNA, but they pose a real diagnostic challenge. According to our analyses, such nodules are classified into category IV of BSRTC ten times more often than non-HC nodules and into category III more than twice as often as non-HC nodules. Slightly lower differences are observed in the case of category V [4]. Similar data were shown by Yazgan et al. [5]. Difficulties in the identification of optimal clinical management of patients with HC nodules are deepened by the fact that many published reports lack the data on the RoM of these nodules in relation to the FNA outcome category. Furthermore, even when such data are presented, the estimated risk of malignancy in HC nodules with equivocal cytology varies significantly between the reporting centers [4,5,6]. It is not only a consequence of differences in the incidence of malignant and benign thyroid nodules related to iodine supply but also a result of inconsistent classification of smears to indeterminate categories of the BSRTC. In particular, category III is characterized by low reproducibility [15], and category IV is the reason for some incoherence as it may be variably defined depending on the application of its recent modification [3]. At our center, the risk of malignancy of HC nodules classified into categories III or IV is low and it does not exceed 15%. This is the consequence of the epidemiological situation of our population (a post-endemic population) and the conservative attitude to classification of nodules into category IV. It is worthy of note that our pathologists avoid misinterpretation of PTC cells as HC. All these factors lead to a relatively low percentage of PTC among cancers, especially in nodules of category IV. Given the RoM does not exceed 15%, it is important to try and limit the number of thyroid surgeries performed for nodules of category III, but also of category IV.

Unfortunately, our analyses show that in the group of nodules of equivocal cytology with HC present in a smear, the EU-TIRADS offers substantial diagnostic efficacy (as measured by AUC) only in the case of nodules of category V of the BSRTC. Malignant HC nodules of category V correspond most often to PTC (over 85% in our material). In the case of categories III and IV, it is FTC and HTC that dominate malignant HC nodules. Ultrasound images of these cancers differ from the images of PTC, which is a cause of lower efficacy of U-RSS [13,14,16]. This is especially visible in category IV in the group of HC nodules, where the percentage of PTC was under 30%. In the case of category III, there were differences in the efficacy of EU-TIRADS between the HC group (37.5% PTC among cancers) and non-HC group (63.0% PTC among cancers). In the non-HC group, the classification of a nodule of category III of BSRTC into the intermediate risk category of EU-TIRADS caused a five-fold increase in its malignancy odds. If such a nodule was classified into the high-risk category, its RoM increased 2.4 fold from its initial FNA-RoM. In the HC group, only classification of a nodule into the high-risk category of EU-TIRADS increased its malignancy odds four-fold, but neither that increase nor the increase in RoM from its FNA-RoM were significant. With the cut-off value set at the high-risk category of EU-TIRADS, the sensitivity in the group of HC nodules was markedly lower than in the non-HC group. When the cut-off value was set at the intermediate risk category, the differences in SEN between HC and non-HC groups decreased, but the increase in sensitivity was still unsatisfactory for HC nodules with indeterminate cytology. Even if that category of EU-TIRADS was to be interpreted as an indication for surgical treatment, 25% cancers in category III and 21.4% in category IV of BSRTC would still be left without treatment in the HC group. In the non-HC group, the same cut-off value (the intermediate risk) would lead to surgical treatment of all cancers of category III of BSTRC and 92.6% of cancers of category IV, while the reduction in the number of performed surgeries would be similar in both groups—about 30%. Setting the cut-off value at the low risk category would result in necessity of surgery in over 99% of HC nodules with indeterminate cytology.

The differences in the images of HC nodules of categories III and IV of BSRTC (e.g., irregular margins have been found as a significant US malignancy feature in category III, but not in category IV) may be a cause of contradictory results on the efficacy of US examination obtained from the joint assessment of HC nodules with indeterminate cytology. The majority of studies concern cases of cytologically suspected HN (currently SHCT), but they often include a period before the introduction of Bethesda classification, and thus before establishing category III. In some cases, the reports do not consider FNA outcomes, but they are focused on ultrasonographic differentiation between histopathologically confirmed HTC and HTA [17,18,19,20,21,22,23]. Maizlin et al. (2008) showed that such nodules displayed a wide spectrum of sonographic appearances and that ultrasonography did not help in distinguishing benign and malignant HN [18]. Kim et al. (2010) reported that hypoechogenicity, irregular margin and calcification were not malignancy predictive factors [19]. Lee et al. (2013) showed that hypoechogenicity was of importance in predicting malignancy of such nodules while Parikh et al. (2013) suggested that hypoechogenicity and hyperechogenicity were significantly associated with malignancy [20,21]. Lee at al. (2010) found that the halo sign and combined intra- and perinodular vascularity should be considered as “indeterminate” sonographic findings of HN that require more intensive diagnostic investigations or even surgical intervention [22]. Ito et al. (2016) reported that the presence of a round and hypoechoic solid nodule or a solid nodule with an irregular border or psammoma calcification spoke in favor of the surgical treatment [23]. Those patterns are similar to categories four and five of EU-TIRADS.

According to our knowledge, the EU-TIRADS system in itself has not been analyzed for its usefulness in the diagnostics of HC nodules. A number of studies that confirmed the discrimination value of EU-TIRADS were based on a material in which PTC dominated cancers and FTC constituted only several percent of all malignant nodules and there was usually no information on the frequency of HTC [24,25,26,27,28,29]. The only published study dedicated to the assessment of the performance of U-RSS in nodules with predominant HC referred to ACR-TIRADS. Santana et al. (2019) did not confirm the usefulness of that system in distinguishing benign and malignant HN [30]. They evaluated 24 HTC and 27 HTA and indicated that similar percentages of carcinomas and adenomas (58% and 48%) were classified as ACR-TIRADS categories four or five. In our study, the threshold level set at category four of EU-TIRADS (intermediate risk) gave the higher sensitivity (82.8%) but lower specificity (24.8% vs. 52% in Santana’s group report). Chaigneau et al. (2015) assessed the usefulness of French TIRADS (which resembles the current EU-TIRADS) in the group of nodules of categories III-V of BSRTC, where diagnoses of HN constituted 26% of category IV nodules, but that group was not analyzed separately. The authors did not find that system suitable to guide subsequent management of patients with indeterminate cytology [31].

In other studies on the usefulness of U-RSS in the diagnostics of nodules with equivocal cytology, there was no separated analysis of HC nodules. The EU-TIRADS was evaluated by Piccardo et al. (2020) [32]. The authors analyzed nodules with indeterminate cytology classified into category TIR3A or TIR3B according to the Italian consensus for the classification and reporting of thyroid cytology and they found that high-risk EU-TIRADS features were significantly associated with malignancy only in the TIR3B subgroup. That category corresponds to category IV according to the BSRTC after its modification in 2017, which means it includes cases with “mild/focal nuclear atypia”. With such a modification, more cancers correspond to PTC than in the original version of BSRTC category IV and more than in category TIR3A, as both those categories are characterized by architectural changes. Several studies concerned K-TIRADS and ATA guideline systems, but they gave contradictory results. Some authors indicated that they made it possible to exclude malignancy in indeterminate thyroid nodules [33], while others suggested their usefulness to predict malignancy in FLUS/AUS nodules [34,35,36] or only in AUS nodules [37,38]. Hong et al. (2017) reported that a high-suspicion US pattern in the K-TIRADS system significantly increased the malignancy risk of FLUS/AUS nodules but not of SFN/SHCT nodules [39]. Those results are concordant with our previous study that compared six systems, including K-TIRADS, but did not assess HC and non-HC nodules separately [13]. Similarly, Yang et al. (2020) did not find K-TIRADS, ATA guidelines and ACR-TIRADS to be useful for the RoM assessment in nodules of category IV [40] and Sahli et al. (2018) found that ACR-TIRADS was a poor predictor of final surgical pathology among cytologically indeterminate and genetically-suspicious nodules [41]. On the other hand, there are also reports indicating the efficacy of ACR-TIRADS and ATA-guidelines in both categories of indeterminate nodules [42,43]. Trimboli et al. (2017) compared the usefulness of four U-RSS (ATA guidelines, AACE/ACE/AME guidelines, ACR-TIRADS and British Thyroid Association (BTA) system) in a sample of 101 nodules with indeterminate cytology and relatively low percentage of PTC among cancers (57%) and showed that U-RSS had suboptimal diagnostic accuracy for making clinical decisions [44].

The limitation of our study is the low number of cancers in the FLUS/AUS and SFN/SHCT subgroups, but this number reflects the low risk of malignancy in such nodules in a population that has been exposed to iodine deficiency. The way of selecting nodules for analysis on the basis of the postoperative histopathological examination is another disadvantage; however, it is the only way to verify the final diagnosis that gives certainty in the case of nodules with equivocal cytology. The advantage of our study is performing US malignancy feature evaluation directly prior to biopsy. Therefore, the result of FNA has not influenced the evaluation.

## 5. Conclusions

In conclusion, the present analysis is one of few concerning the evaluation of usefulness of U-RSS in diagnostics of nodules with the predominance of HC and it is the first one relating to EU-TIRADS. The analysis indicates that the system does not allow clinical decisions to be made in patients with HC nodules of indeterminate cytology, particularly those classified into category IV of BSRTC. In the case of category III in the Bethesda classification, the EU-TIRADS shows a higher potential but it relates mainly to non-HC nodules. We believe this is a consequence of the higher percentage of PTC among cancers in that group of nodules, as U-RSS have been optimized to achieve maximum efficiency in revealing this very type of thyroid cancers. Paradoxically, the lower the number of errors made during cytological interpretation by mistaking PTC cells for HC, the lower the efficacy of U-RSS. The current study, as well as our previous reports, suggests that when the percentage of PTC among cancers is <40%, the efficiency of U-RSS is low and only becomes acceptable when that percentage exceeds 60%. Thus, the knowledge of the relative incidence of PTC in each BSRTC category is a prerequisite of any prudent interpretation of the results of U-RSS evaluation in the case of cytologically equivocal nodules. It should be emphasized that these incidences are specific not only to particular populations but also to particular cytological centers because the reproducibility of diagnoses of equivocal Bethesda categories is quite low. It is only possible to decide whether any particular TIRADS category indicates the necessity of surgical treatment if data on the malignancy risk and the incidence of particular types of carcinomas specific for a given equivocal Bethesda category are considered. It seems that the inclusion of additional data from molecular studies could help in a more precise identification of nodules that ought to be excised, especially if the molecular tests specific for Hürthle cell nodules could be identified.

## Figures and Tables

**Table 1 jcm-09-03410-t001:** Demographic data of the patients and the percentage of cancers revealed in the Hürthle cell (HC) and non-HC nodules.

Parameter	HC			Non-HC			*p*
Number of nodules	162			378			
Number of patients	138			347			
Age—mean ± SD (years)	56.3 ± 13.9			53.0 ± 13.9			<0.05
No/% of males	10/7.2			42/12.1			NS
Volume of nodules mean ± SD (cm^3^)	6.41 ± 15.0			6.00 ±11.9			NS
No/% of cancers	29/17.9			59/15.6			NS
No/% of PTCs among cancers	13/44.8			44/74.6			<0.01
Other cancers (No/%)	FTC (4/13.8), HTC (11/37.9) MTC (1/3.4)			FTC (10/16.9), HTC (1/1.7), MTC (2/3.4), AC (1/1.7), ANG (1/1.7)			HTC: <0.0001
	**category of BSRTC**						
	**III**	**IV**	**V**	**III**	**IV**	**V**	
No/% of cancers among nodules in each category of BSRTC/*p*	8/15.1	14/14.4	7/58.3	27/9.8	5/7.1	27/84.4	NS
	HC: <0.005: V vs. III & IV			non-HC: <0.0001: V vs. III & IV			
No/% of PTCs among cancers in each category of BSRTC/*p*	3/37.5	4/28.6	6/85.7	17/63.0	2/40.0	25/92.6	NS
	HC: <0.05 V vs. IV			non-HC: <0.05 V vs. III & IV			

BSRTC—Bethesda System for Reporting Thyroid Cytopathology, PTC—papillary thyroid carcinoma, MTC—medullary thyroid carcinoma, FTC—follicular thyroid carcinoma, HTC—Hürthle cell thyroid carcinoma, AC—anaplastic carcinoma, ST—secondary tumor, ANG—angiosarcoma.

**Table 2 jcm-09-03410-t002:** Comparison of the incidence of sonographic features in HC and non-HC nodules in relation to the histopathological outcome: benign lesion vs. thyroid malignancy. Results of univariate logistic regression analysis in both groups.

Sonographic Feature *		HC Nodules			Non-HC Nodules			
	Ben. (133) No/%	Mal. (29) No/%	*p*	OR CI95% *p*	Ben. (319) No/%	Mal. (59) No/%	*p*	OR CI95% *p*
marked hypoechogenicity	6/4.5	7/24.1	<0.001	6.7 (2.1–21.9) 0.002	28/8.8	14/23.7	<0.001	3.2 (1.6–6.6) 0.001
hypoechogenicity	95/71.4	24/82.8	NS	1.9 (0.7–5.4) 0.216	223/69.9	49/83.1	<0.05	2.1 (1.0–4.3) 0.004
solid echostructure	121/91.0	25/86.2	NS	0.6 (0.2–2.1) 0.439	243/76.2	52/88.1	<0.05	2.3 (1.0–5.3) 0.005
more solid than cystic echostructure	131/98.5	28/96.5	NS	0.4 (0.1–4.9) 0.494	271/84.9	58/98.3	<0.01	4.9 (0.7–37.1) <0.121
spongiform echostructure	2/1.5	0/0/0	NS	0.0 (0.0–) 0.998	11/3.4	0/0.0	NS	0.0 (0.0–) 0.997
suspicious shape	15/11.3	7/24.1	NS	2.5 (0.9–6.8) 0.074	34/10.7	11/18.6	NS	1.9 (0.9–4.0) 0.086
irregular margins	3/2.3	5/17.2	<0.001	9.0 (2.0–40.3) 0.004	14/14.4	18/30.5	<0.0001	9.6 (4.4–20.7) <0.0001
microcalcifications	5/3.8	5/17.2	<0.01	5.3 (1.4–19.8) 0.013	12/3.8	9/15.3	<0.0005	4.6 (1.8–11.5) 0.001
macrocalcifications	3/2.3	3/10.3	NS	5.0 (1.0–26.2) 0.057	23/7.2	8/13,6	NS	2.0 (0.9–4.8) 0.108
rim calcifications	3/2.3	2/6.9	NS	3.2 (0.5–20.1) 0.213	10/3,1	3/5,1	NS	1.4 (0.4–6.2) 0.454
pathologicalvascularization	30/22.6	9/31.0	NS	1.5 (0.6–3.7) 0.3365	77/24.1	16/27.1	NS	1.2 (0.6–2.2) 0.626

*—none of the nodules of both groups presented a pure cystic echostructure. Ben.—benign lesion in histopathological outcome. Mal.—thyroid malignancy in histopathological outcome.

**Table 3 jcm-09-03410-t003:** Distribution of benign and malignant nodules (and their types) between particular categories of EU-TIRADS, the comparison of expected T-RoM with calculated T-RoM. (EU-TIRADS category corresponding to the lack of nodules has been omitted).

Category of TIRADS	No/% of Nodules	No/% ben. nod	No/% mal. nod.	Calc. T-RoM	Expec. T-RoM	No/% of PTC	No/% of FTC	No/% of HTC	No/% of MTC	No/% of ATC	No/% of ANG
	HC nodules										
benign	2/1.2	2/1.5	0/0	0.0	0						
low risk	36/22.2	31/23.3	5/17.2	13.9	2–4	2/15.4	1/25.0	2/18.2			
intermediate risk	83/51.2	72/54.1	11/37.9	13.3	6–17	3/23.1	2/50.0	5/45.4	1/100.0		
high risk	41/25.3	28/21.1	13/44.8	31.7	26–87	8/61.5	1/25.0	4/36.4			
	**non-HC nodules**										
benign	10/2.6	10/3.1	0/0	0.0	0						
low risk	86/22.8	80/25.1	6/10.2	7.0	2–4	5/11.4	1/10.0				
intermediate risk	175/46.3	158/49.5	17/28.8	9.7	6–17	11/25.0	5/50.0	1/100.0			
high risk	107/28.3	71/22.3	36/61.0	33.6	26–87	28/63.6	4/40.0		2/100.0	1/100.0	1/100.0

PTC—papillary thyroid carcinoma; MTC—medullary thyroid carcinoma; FTC—follicular thyroid carcinoma; HTC—Hürthle cell thyroid carcinoma; AC—anaplastic carcinoma; ST—secondary tumor; ANG—angiosarcoma; T-RoM—risk of malignancy in a nodule related to its TIRADS category.

**Table 4 jcm-09-03410-t004:** Data on the diagnostic efficacy of EU-TIRADS in relation to the category of Bethesda System for Reporting Thyroid Cytopathology (BSRTC) in examined groups of nodules.

	Category of BSRTC							
	All Categories		III		IV		V	
	HC	Non-HC	HC	Non-HC	HC	Non-HC	HC	Non-HC
AUC	0.621	0.711	0.628	0.716	0.582	0.674	0.829	0.859
95% Cl	0.502–0.739	0.638–0.784	0.396–0.860	0.616–0.816	0.404–0.760	0.463–0.884	0.591–1.0	0.726–0.992
*p*	<0.05	<0.0001	0.280	<0.0001	0.366	0.105	0.007	<0.0001
	**EU-TIRADS threshold category: high risk (5)**							
SEN	44.8	61.0	37.5	55.6	50.0	60.0	42.9	66.7
SPC	78.9	77.7	88.9	80.3	72.3	66.2	100.0	100.0
ACC	72.8	75.1	81.1	77.9	69.1	65.7	66.7	71.9
PPV	31.7	33.6	37.5	23.4	23.3	12.0	100.0	100.0
NPV	86.8	91.5	88.9	94.3	89.6	95.6	55.6	35.7
OR	3.0	5.5	4.8	5.1	2.5	2.9	-	-
95% Cl	1.3–7.1	3.0–9.8	0.9–26.4	2.2–11.6	0.8–8.3	0.5–18.9		
*p*	0.009	< 0.0001	0.072	0.0001	0.103	0.257		
	**EU-TIRADS threshold category:** **intermediate risk (4)**							
SEN	82.8	89.8	75.0	92.6	78.6	100.0	100.0	85.2
SPC	24.8	28.2	33.3	29.3	19.3	21.5	40.0	60.0
ACC	35.2	37.8	39.6	35.5	27.8	27.1	75.0	81.3
PPV	19.4	1.8	16.7	12.4	14.1	8.9	70.0	92.0
NPV	86.8	93.8	88.2	97.3	84.2	100.0	100.0	42.9
OR	1.6	3.5	1.5	5.2	0.9	-	-	8.6
95% Cl	0.6–4.5	1.4–8.4	0.3–8.3	1.2–22.4	0.2–3.5			1.1–69.1
*p*	0.386	0.005	0.643	0.028	0.851			0.042

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
