# Peer review of "Usability of EU-TIRADS in the Diagnostics of Hürthle Cell Thyroid Nodules with Equivocal Cytology"

_jcm, 2020, doi:10.3390/jcm9113410_

Round 1
Reviewer 1 Report
Usability of EU-TIRADS in the diagnostics of Hürthle cell thyroid nodules with equivocal cytology
Dear Authors,
I found your study clear and well written. I only have some minor comments.
The discussion is quite long (but detailed). Still I miss some future directions how to improve ‘clinical decision making in patients with cytologically equivocal HC nodules.’ It is clear from this study that EU-TRIADS fails to do so. Would molecular typing of micro-dissected cells, although more expensive, aid? This should be discussed.
Minor comment: personally I prefer the use of the chemically correct name ‘iodide’ instead of ‘iodine.’ Thyroid epithelial cells express NIS (sodium iodide symporter) and thyroid cancer patients are frequently treated with 131I- (iodide). Please change iodine into iodide.
Author Response
We sincerely thank for the opinion that the paper is clear and well-written.
- The discussion is quite long (but detailed). Still I miss some future directions how to improve ‘clinical decision making in patients with cytologically equivocal HC nodules.’ It is clear from this study that EU-TRIADS fails to do so. Would molecular typing of micro-dissected cells, although more expensive, aid? This should be discussed.
The discussion has been shortened in the part concerning the analysis of particular US malignancy risk features (lines 308-341) as well as in the part concerning the evaluation of U-RSS systems other than EU-TIRADS (lines 378-385).
The Conclusions section has been expanded. Its revised version includes major factors which have an impact on the clinical interpretation of the results of U-RSS evaluation, like the risk of malignancy of a nodule related to its BSTRC category and the incidence of PTC among cancers in nodules of that category. We have suggested the values of PTC incidence at which the efficacy of EU-TIRADS in making decisions on the necessity of surgical treatment is acceptable. It has been pointed out that this efficacy depends on the configuration of FNA-RoM and PTC incidence among cancers in a particular diagnostic center. Molecular tests have been mentioned as another potential method for improving the diagnostics of HC nodules.
- Minor comment: personally I prefer the use of the chemically correct name ‘iodide’ instead of ‘iodine.’ Thyroid epithelial cells express NIS (sodium iodide symporter) and thyroid cancer patients are frequently treated with 131I- (iodide). Please change iodine into iodide
Unfortunately, we cannot agree with the use of the term ‘iodide’ instead of ‘iodine’. Both terms are correct but they have different meanings: ‘iodine’ is a name of the chemical element while ‘iodide’ refers to one of several oxidation states in which iodine occurs (like iodide anion [I-] in sodium iodide or potassium iodide). There are other iodine oxidation states, e.g. iodate (IO3-). Therefore, all iodides are iodine but not vice versa. Thus, when speaking about deficiency of the chemical element in humans we should definitely use the term ‘iodine’. Accordingly, the worldwide control of iodine deficiency disorders is a goal of the Iodine Global Network organization (former International Council for Control of Iodine Deficiency Disorders). The abbreviation NIS does indeed stand for sodium/iodide symporter and not sodium/iodine symporter, but it’s because this protein is able to transport iodine through the cellular membrane only in the form of iodide anions.
Reviewer 2 Report
The manuscript by Klencka et al. has tried to properly address the use of EU-TIRADS for the diagnostics of Hürthle cell thyroid nodules with respect to equivocal cytology.
Some comments for improvement of the manuscript are:
1. The conclusion is too succinct, needs to be expanded. The future directions and prospects can be incorporated.
2. There are several typographical and grammatical errors throughout the manuscript which requires serious attention.
3. In the abstract, the terms such as RoM should be properly explained.
4. A schematic diagram can be added to sum up the findings and further research/investigations required.
5. The discussion part is very long, and should be shortened only to include the relevant and essential information.
Author Response
- The conclusion is too succinct, needs to be expanded. The future directions and prospects can be incorporated.
The Conclusions section has been expanded. Its revised version includes major factors which have an impact on the clinical interpretation of the results of U-RSS evaluation, like the risk of malignancy of a nodule related to its BSTRC category and the incidence of PTC among cancers in nodules of that category. We have suggested the values of PTC incidence at which the efficacy of EU-TIRADS in making decisions on the necessity of surgical treatment is acceptable. It has been pointed out that this efficacy depends on the configuration of FNA-RoM and PTC incidence among cancers in a particular diagnostic center. Molecular tests have been mentioned as another potential method for improving the diagnostics of HC nodules.
- There are several typographical and grammatical errors throughout the manuscript which requires serious attention.
The manuscript has been thoroughly checked for that type of errors and several corrections have been made (at lines 75, 121, 269, 320, 347 , 368, 397 of the original version and line 324 of the revised version)
- In the abstract, the terms such as RoM should be properly explained.
The explanation of the abbreviation ‘RoM’ has been inserted into the abstract as suggested.
- A schematic diagram can be added to sum up the findings and further research/investigations required.
This suggestion was considered in detail but we came to the conclusion that such a diagram could wrongly imply that there is only one optimal way to deal with HC nodules in relation to the EU-TIRADS and BSRTC categories. Actually, clinical decisions should be made with consideration of some diagnostic center-specific features like FNA-RoM related to particular BSRTC categories and the incidence of PTC among caners in each of these categories. Both these factors may differ significantly between various centers and, consequently, different EU-TIRADS categories may be the optimal cut-off value for surgical treatment. This issue has been underlined in the Conclusions.
- The discussion part is very long, and should be shortened only to include the relevant and essential information.
The discussion has been shortened in the part concerning the analysis of particular US malignancy risk features (lines 308-341) as well as in the part concerning the evaluation of U-RSS systems other than EU-TIRADS (lines 378-385).
Round 2
Reviewer 2 Report
The manuscript has now been modified appropriately based on the comments/suggestions provided.